# Systemic Cancer Therapy Does Not Significantly Impact Early Vaccine-Elicited SARS-CoV-2 Immunity in Patients with Solid Tumors

**DOI:** 10.3390/vaccines10050738

**Published:** 2022-05-09

**Authors:** Adam T. Waickman, Joseph Lu, Corey Chase, Hengsheng Fang, Erinn McDowell, Erin Bingham, Jeffrey Bogart, Stephen Graziano, Stephen J. Thomas, Teresa Gentile

**Affiliations:** 1Department of Microbiology and Immunology, State University of New York Upstate Medical University, Syracuse, NY 13210, USA; luq@upstate.edu (J.L.); fangh@upstate.edu (H.F.); thomstep@upstate.edu (S.J.T.); 2Institute for Global Health and Translational Sciences, State University of New York Upstate Medical University, Syracuse, NY 13210, USA; 3Department of Hematology and Oncology, State University of New York Upstate Medical University, Syracuse, NY 13210, USA; chasec@upstate.edu (C.C.); mcdowele@upstate.edu (E.M.); binghame@upstate.edu (E.B.); grazians@upstate.edu (S.G.); gentilet@upstate.edu (T.G.); 4Department of Medical Oncology, State University of New York Upstate Medical University, Syracuse, NY 13210, USA; bogartj@upstate.edu

**Keywords:** cancer therapy, COVID-19, vaccine-elicited immunity, antibody titer, T cell response

## Abstract

mRNA vaccines have been shown to be safe and effective in individuals with cancer. It is unclear, however, if systemic anti-cancer therapy impacts the coordinated cellular and humoral immune responses elicited by SARS-CoV-2 mRNA vaccines. To fill this knowledge gap, we assessed SARS-CoV-2 mRNA vaccine-elicited immunity in a cohort of patients with advanced solid tumors either under observation or receiving systemic anti-cancer therapy. This analysis revealed that SARS-CoV-2 mRNA vaccine-elicited cellular and humoral immunity was not significantly different in individuals with cancer receiving systemic anti-cancer therapy relative to individuals under observation. Furthermore, even though some patients exhibited suboptimal antibody titers after vaccination, SARS-CoV-2 specific cellular immune responses were still detected. These data suggest that antibody titers offer an incomplete picture of vaccine-elicited SARS-CoV-2 immunity in cancer patients undergoing active systemic anti-cancer therapy, and that vaccine-elicited cellular immunity exists even in the absence of significant quantities of SARS-CoV-2 specific antibodies.

## 1. Introduction

Individuals with cancer are at significantly higher risk of unfavorable outcomes following SARS-CoV-2 infection relative to the general population [1,2,3,4,5,6]. This manifests as an increased likelihood of hospitalization, intensive care unit admission, intubation, and death due to coronavirus disease 2019 (COVID-19) [1,2,3,4,5,6]. This is thought to be at least partially attributable to common comorbidities observed in individuals with cancer such as advanced age, immunosenescence, and basal lung inflammation that can potentiate and accelerate SARS-CoV-2 pathogenesis [7]. Furthermore, many first-line cancer therapies—such as radiation, immune checkpoint inhibitors, and systemic chemotherapy—are associated with increased basal inflammation, acute immunosuppression, and blunted vaccine immunogenicity [8,9]. Accordingly, patients with cancer were among the first individuals prioritized to receive a SARS-CoV-2 vaccine by the United States Centers for Disease Control and Prevention early during the pandemic, and were also among the first authorized and recommended to receive a booster vaccine in August 2021.

Multiple studies have now shown that SARS-CoV-2 mRNA vaccines—such as BNT162b2 (BioNTech; Pfizer ) and mRNA-1273 (Moderna)—are generally well tolerated, immunogenic, and effective in individuals with a history of cancer [10,11]. However, some heterogeneity in SARS-CoV-2 vaccine immunogenicity and efficacy has been observed, especially in individuals with lung cancer, hematological malignancies, and those receiving systemic anti-cancer treatments, such as anti-CD20/BTK inhibitor therapy [12,13,14,15,16]. Furthermore, the majority of the immunogenicity studies performed to date have focused on the quantification of SARS-CoV-2 spike protein-reactive antibodies, leaving unresolved the contribution of cell mediated immunity (CMI) to the vaccine-elicited immune profile. Therefore, while SARS-CoV-2 mRNA vaccines appear to be generally effective in individuals with cancer, the extent to which systemic anti-cancer therapy may impact the coordination of humoral vaccine immunogenicity in this high-risk group remains unclear.

To help fill this knowledge gap we performed a prospective analysis of SARS-CoV-2 mRNA vaccine humoral and cellular immunogenicity in individuals with cancer. This study focused on determining if the levels of SARS-CoV-2 spike RBD specific antibodies or the frequency of SARS-CoV-2 spike-reactive T cells were impacted by systemic anti-cancer therapy, relative to individuals with cancer who were not receiving active therapy at the time of vaccine administration.

## 2. Methods

### 2.1. Study Design

A prospective, single-center observational cohort study was initiated in April 2021 that included patients receiving treatment for solid tumors at the Upstate Cancer Center or its satellite treatment facilities located in Syracuse, NY. Enrollment in the study was restricted to individuals with locally advanced or metastatic solid tumors (excluding hematologic malignancies—lymphoma, myeloma, leukemia) who had received their second dose of either the BNT162b2 mRNA SARS-CoV-2 vaccine (BioNTech; Pfizer) or mRNA-1273 mRNA SARS-CoV-2 vaccine (Moderna) within the previous 8 weeks. PBMCs were obtained following gradient centrifugal separation of peripheral blood collected using Cell Preparation Tubes (CPT; BD Biosciences, Franklin lakes, NJ, USA), and serum isolated using Serum Separation Tubes (SST; BD Biosciences, Franklin lakes, NJ, USA). All samples analyzed in the study were collected between 12 April and 7 July, 2021.

The control group for this study consisted of patients that either (1) had been previously treated for cancer (excluding hematologic malignancies—lymphoma, myeloma, leukemia) without active disease, (2) were receiving hormonal therapy only for breast or prostate cancer, (3) were undergoing radiation therapy, (4) are undergoing radiation therapy and hormonal therapy only for breast or prostate cancer, or (5) undergoing active surveillance for cancer that did not currently require therapy. Information on the type of cancer and the nature of any active therapy was collected from study participants in addition to the date of SARS-CoV-2 mRNA vaccine administration. Participants did not receive financial compensation for participation in the study, and all study activities were approved by the SUNY Upstate Institutional Review Board.

### 2.2. SARS-CoV-2 Spike ELISA

SARS-CoV-2 Spike RBD antibody titers were quantified using a sandwich ELISA protocol. In brief, 96 well NUNC MaxSorb flat-bottom plates were coated with 1 μg/mL of recombinant SARS-CoV-2 Wuhan-Hu-1 spike RBD protein (Sino Biological, Beijing, China, cat. 40592-V08B) diluted in sterile 1x PBS. Plates were washed and blocked for 30 min at RT with 0.25% BSA + 1% Normal Goat Serum in 0.1% PBST after overnight incubation at 4 °C. Serum samples were heat-inactivated for 30 min at 56 °C and serially diluted 4-fold, eight times, starting at 1:200 prior to incubation for 2 h at RT on the blocked plates. Plates were washed and antigen-specific antibody binding was detected using anti-human IgG HRP (MilliporeSigma, St. Louis, MO, USA, cat. SAB3701362), or anti-human IgM HRP (SeraCare, Milford, MA, USA, Cat. 5220-0328). Secondary antibody binding was quantified using the Pierce TMB Substrate Kit (Thermo, cat. 34021, Waltham, MA, USA) and Synergy HT plate reader (BioTek, Winooski, VT, USA). Antibody binding data were analyzed by nonlinear regression (one site specific binding with Hill slope) of background-subtracted OD450 values to determine EC_50_ titers, reported as Kd values, in GraphPad Prism 9.1.0 (GraphPad Software, La Jolla, CA, USA).

### 2.3. IFN-γ ELISPOT

Cryopreserved PBMC were thawed, washed twice, and placed in complete cell culture media: RPMI 1640 medium (Corning, Tewksbury, MA, USA) supplemented with 10% heat-inactivated fetal calf serum (Corning, 35-010-CV), L-glutamine (Lonza, Basel, Switzerland), and Penicillin/Streptomycin (Gibco, Waltham, MA, USA). Cellular viability was assessed by trypan blue exclusion and cells were resuspended at a concentration of 5 × 10^6^/mL and rested overnight at 37 °C. After resting, viable PBMC were washed, counted, and resuspended at a concentration of 1 × 10^6^/mL in complete cell culture media. Next, 100 mL of this cell suspension was mixed with 100 mL of a SARS-CoV-2 Spike peptide pool (BEI, cat. NR52402) diluted to a final concentration of 1 μg/mL/peptide (DMSO concentration 0.5%) in complete cell culture media. This cell and peptide mixture was loaded onto a 96-well PVDF plate coated with anti-IFN-γ (3420-2HW-Plus, Mabtech, Nacka, Sweden) and cultured overnight. Controls for each donor included 0.5% DMSO alone (negative) and anti-CD3 (positive). After overnight incubation, the ELISPOT plates were washed and stained with anti-IFN-γ-biotin followed by streptavidin-conjugated HRP (3420-2HW-Plus, Mabtech). Plates were developed using TMB substrate and read using a CTL-ImmunoSpot^®^ S6 Analyzer (Cellular Technology Limited, Shaker Heights, OH, USA). All peptide pools were tested in duplicate, while the negative control (DMSO only) was run in triplicate. The adjusted mean was reported as the mean of the duplicate experimental wells after subtracting the mean value of the negative control wells. All data were normalized based on the number of cells plated per well and are presented herein as SFC/10^6^ PBMC.

### 2.4. Statistical Analysis

Statistical analyses were performed using GraphPad Prism v9 Software (GraphPad Software, La Jolla, CA, USA). A *p*-value < 0.05 was considered significant.

## 3. Results

### 3.1. Study Design and Demographics

A total of 86 participants were enrolled in this study, with 31 individuals in the control group and 55 receiving active systemic anti-cancer therapy at the time of SARS-CoV-2 mRNA vaccination (Table 1). Age, sex, and the type of SARS-CoV-2 mRNA vaccine received were equivalent between the two study arms (Table 1). Within the systemic treatment group, at the time of SARS-CoV-2 mRNA vaccine administration, 14/55 patients were receiving chemotherapy alone, 21/55 were receiving immunotherapy alone, 8/55 were receiving chemotherapy and immunotherapy, and 12/55 were receiving concurrent chemotherapy, immunotherapy, and radiation therapy (Table 2).

### 3.2. SARS-CoV-2 Vaccine Immunogenicity

Upon study enrollment, all subjects exhibited a SARS-CoV-2 Spike RBD-specific IgG EC_50_ titer of >1:200 (Figure 1A). The mean IgG titer observed in the control and treatment arms of the study were 1:67,063 and 1:48,182, respectively. Consistent with the timing post vaccination, only modest SARS-CoV-2 Spike RBD-specific IgM responses were observed in all subjects (Figure 1B). No statistically significant difference was observed for anti-SARS-CoV-2 Spike RBD IgG (*p* = 0.4452) or IgM (*p* = 0.3562) titers between the control and treatment arms of the study (Figure 1A,B). Similarly, 79 of the 86 study participants exhibited a SARS-CoV-2 Spike-specific T cell response upon enrollment, defined as >50 SFC/10^6^ PBMC. No statistically significant difference in the SARS-CoV-2 Spike specific T cell response was observed between the control and treatment arms of the study (Figure 1C).

A statistically significant correlation was observed between SARS-CoV-2 Spike RBD IgG titers and the frequency of Spike-reactive T cells quantified by IFN-g ELISPOT (Figure 1D). However, it was noted—with the exception of one individual—that those individuals with the lowest IgG antibody titers still exhibited a Spike-reactive T cell response above our positivity threshold of 50 SFC/10^6^ PBMC (Figure 1D, Table 3).

## 4. Discussion

In this study, we observed that neither SARS-CoV-2 spike antibody titers nor T cell responses following COVID-19 mRNA vaccination were significantly reduced in individuals with advanced cancer receiving systemic anti-cancer therapy, relative to individuals with cancer not receiving active systemic therapy. Furthermore, while SARS-CoV-2 spike-specific antibody and T cell responses exhibited a significant degree of correlation across both arms of our study, with one exception, those individuals with the lowest antibody titers following vaccination still exhibited a positive SARS-CoV-2 spike-specific T cell response. These results highlight the importance of considering both humoral and cellular immunity following vaccination, and suggest that SARS-CoV-2-specific immunity may still be present in individuals with low antibody titers.

The development of SARS-CoV-2-specific cellular immunity has the potential to play a significant role in providing durable protection against severe COVID-19 in both healthy individuals and those with cancer. SARS-CoV-2 specific memory T cells are readily detectable in circulation after both natural SARS-CoV-2 infection as well as following vaccination with either of the two mRNA vaccine products described in this study [17]. Furthermore, the presence of pre-existing/cross-reactive SARS-CoV-2 specific T cells in the absence of vaccination is associated with protection from SARS-CoV-2 infection [18,19]. Accordingly, understanding to breadth and function potential of SARS-CoV-2 reactive memory T cells in immunologically vulnerable individuals may provide a better understanding of vaccine elected immunity and durable protection from symptomatic SARS-CoV-2 infection than antibody titers alone.

While infection- and vaccine-elicited antibody responses against recently emergent SARS-CoV-2 variants have been shown to be somewhat blunted relative to earlier SARS-CoV-2 strains, T cell responses against these same variants have generally been well maintained [20,21,22]. Booster vaccinations have been shown to further expand the cross-reactivity and frequency of SARS-CoV-2 specific antibodies and cellular immune responses [23]. However, it has yet to be determined if this is also observed in individuals with cancer with or without concurrent anti-cancer therapy.

While the results presented herein are consistent with other reports describing overall high levels of SARS-CoV-2 vaccine immunogenicity in individuals with solid tumors even in the presence of systemic anti-cancer therapy [11,24], there are limitations of this study that must be acknowledged. First, only a single time point was analyzed from each subject, leaving open the possibility that the durability of SARS-CoV-2 vaccination may differ between the study group arms. Second, the relatively small sample size and the unknown prior SARS-CoV-2 infection history may limit the statistical power of the analysis presented herein. Finally, no comparison was made in this analysis to individuals without cancer. However, these data broadly support and reinforce the immunogenicity of SARS-CoV-2 mRNA vaccines in individuals with cancer.

## Figures and Tables

**Figure 1 vaccines-10-00738-f001:**
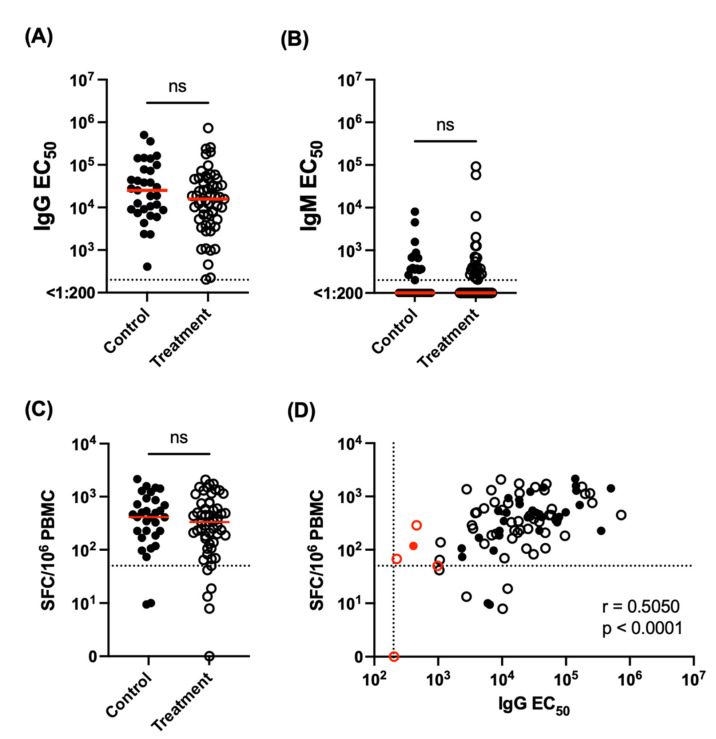
Quantification of SARS-CoV-2-specific humoral and cellular immunity in cancer patients undergoing systemic therapy. (**A**) SARS-CoV-2 spike RBD IgG titers as assessed by ELISA. Unpaired t test. Dotted line indicates assay positive cutoff (EC_50_ > 200). (**B**) SARS-CoV-2 spike RBD IgM titers as assessed by ELISA. Unpaired *t* test. Dotted line indicates assay positive cutoff (EC_50_ > 200). (**C**) SARS-CoV-2 spike specific cellular immunity as quantify by IFN-g ELISPOT. Dotted line indicates assay positive threshold of 50 SFC/10^6^ PBMC. (**D**) Correlation between spike RBD IgG antibody titers and total spike cellular immune response. Individuals with the lowest IgG titers highlighted in red. Filled symbol = control group. Open symbol = treatment group. Spearman correlation analysis. Dotted lines indicate positive cutoff thresholds for each assay.

**Table 1 vaccines-10-00738-t001:** Study population demographic.

	Control (*n* = 31)	Treatment (*n* = 55)
Median Age (range)	64 (50–83)	66 (41–85)
Sex		
Male	54.8% (17/31)	49.1% (27/55)
Female	45.2% (14/31)	50.9% (28/55)
Vaccine type		
Pfizer	54.8% (17/31)	60.0% (33/55)
Moderna	45.2% (14/31)	40.0% (22/55)
Primary cancer location		
Anus	0% (0/31)	5.5% (3/55)
Bile Duct	0% (0/31)	1.8% (1/55)
Breast	25.8% (8/31)	1.8% (1/55)
Colon	0% (0/31)	7.3% (4/55)
Endometrium	3.2% (1/31)	1.8% (1/55)
Esophagus	0% (0/31)	1.8% (1/55)
H&N	9.7% (3/31)	1.8% (1/55)
Liver	0% (0/31)	3.6% (2/55)
Lung	16.1% (5/31)	58.2% (32/55)
Pancreas	0% (0/31)	10.9% (6/55)
Parotid	3.2% (1/31)	0% (0/55)
Peritoneum	0% (0/31)	1.8% (1/55)
Prostate	38.7% 12/31	0% (0/55)
Rectum	3.2% (1/31)	0% (0/55)
Tongue	0% (0/31)	1.8% (1/55)
Vulva	0% (0/31)	1.8% (1/55)
Stage		
0	3.2% (1/31)	0% (0/55)
I	25.8% (8/31)	1.8% (1/55)
II	41.9% (13/31)	7.3% (4/55)
III	19.4% (6/31)	32.7% (18/55)
IV	9.7% (3/31)	58.2% (32/55)

**Table 2 vaccines-10-00738-t002:** Systemic anticancer treatment groups.

Systemic Anticancer Treatment	
Chemotherapy	25.5% (14/55)
Immunotherapy	38.2% (21/55)
Chemotherapy and Immunotherapy	14.5% (8/55)
Chemotherapy, immunotherapy, and radiation	21.8% (12/55)

**Table 3 vaccines-10-00738-t003:** Details on antibody non-responders.

Age	Sex	Primary Tumor Site	Stage	Current Treatment	Vaccine	IgG EC_50_	ELISPOT SFC/10^6^ PBMC
74	F	Lung	IIIA	Durvalumab	Pfizer	1:204	0.00
77	M	Lung	IIIA	Alimta, Keytruda	Pfizer	1:224	67.50
69	M	Prostate	IIIB	N/A	Pfizer	1:409	118.33
57	F	Lung	IV	Pembrolizumab, Pemetrexed	Pfizer	1:459	287.78
67	M	Lung	IVB	Pembrolizumab, Carboplatin, Alimta	Pfizer	1:980	50.00

## Data Availability

The authors declare that all data supporting the findings of this study are available within this article or from the corresponding author upon reasonable request.

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
