# Peer review of "Systemic Cancer Therapy Does Not Significantly Impact Early Vaccine-Elicited SARS-CoV-2 Immunity in Patients with Solid Tumors"

_vaccines, 2022, doi:10.3390/vaccines10050738_

Round 1

Reviewer 1 Report

Waickman et al. reported an interesting observation that cancer patients with low antibody titers following vaccination still exhibited a positive SARS-CoV-2-specific T cell response, and claimed that SARS-CoV-2-specific immunity may be present in cancer patients undergoing therapies even with low antibody titers. The narrative of the manuscript is complete and sound, and the English usage is clear and professional. The authors may want to provide more details in their experimental design and statistical analysis, and provide a clearer picture of why is this research significant and what the readers should learn from these observations. Please find my detailed comments below:

(1) The keywords selected in the keyword section are too vague: “vaccine” does not seem to be the focus of the manuscript. “Vaccine-elicited immunity” may be more accurate. Also, “cancer therapy” may more accurately reflect the focus of the manuscript than “cancer”. Other keywords may be added to better attract readers and give them an overview of the manuscript, including “antibody titer”, “T-cell response”, etc.

(2) In the method or result section, how the samples from patients were collected and treated should be briefly mentioned.

(3) The p values of the Figure 1A and Figure 1B data should be reported.

(4) A statistical analysis paragraph should be included in the method section.

(5) In the introduction or discussion section, the authors may include some literature evidence/citations to highlight that T cells alone are also helpful for protecting against SARS-CoV-2, even if antibody levels are insufficient. The addition of literature evidence will support the argument that it is important to consider both humoral and cellular immunity following vaccination of cancer patients.

(6) Line 121-124: It may be unclear whether all those treatment groups received vaccine administration at the time of receiving therapy. Also, “alone” may be added after “immunotherapy”. I recommend writing like this “within the systemic treatment group, at the time of SARS-CoV-2 mRNA vaccine administration, 14/55 patients were receiving chemotherapy alone, 21/55 were receiving immunotherapy alone, 8/55 were receiving chemotherapy and immunotherapy, and 12/55 were receiving concurrent chemotherapy, immunotherapy, and radiation therapy.”

(7) Line 171-174: these statements seem to be less relevant to the topic of the paper; the authors may further state the significance of this research observation in the discussion section. 

(8) Line 112: why a duplicate run instead of the more commonly used triplicate run was performed?

(9) Figure 1: please provide an explanation of the meaning of the dotted lines in the caption.

Minor points:

(10) Table 1, “58.2% 32/55)”, a parenthesis is missing.

(11) Reference 14, 17, and 18: page numbers are missing.

(12) Line 61, line 79, and line 95: keep the formatting of the subtitle consistent (period vs colon vs nothing).

(13) Table 1: some of the numbers are bold numbers and some of them are not. Please provide an explanation or keep the formatting consistent.

Author Response

(1) The keywords selected in the keyword section are too vague: “vaccine” does not seem to be the focus of the manuscript. “Vaccine-elicited immunity” may be more accurate. Also, “cancer therapy” may more accurately reflect the focus of the manuscript than “cancer”. Other keywords may be added to better attract readers and give them an overview of the manuscript, including “antibody titer”, “T-cell response”, etc.

We thank the reviewer for these suggestions and have modified our manuscript accordingly.

(2) In the method or result section, how the samples from patients were collected and treated should be briefly mentioned.

We have added the requested information to the Materials and Methods section of our revised manuscript

(3) The p values of the Figure 1A and Figure 1B data should be reported.

We have added the indicated p values to the text of our revised manuscript.

(4) A statistical analysis paragraph should be included in the method section.

The suggested paragraph has been added to the Materials and Methods section of our revised manuscript

(5) In the introduction or discussion section, the authors may include some literature evidence/citations to highlight that T cells alone are also helpful for protecting against SARS-CoV-2, even if antibody levels are insufficient. The addition of literature evidence will support the argument that it is important to consider both humoral and cellular immunity following vaccination of cancer patients.

We have added additional language to the introduction section of our manuscript highlighting the proposed role of SARS-CoV-2 reactive T cells in limiting infection and COVID-19 pathogenesis.

(6) Line 121-124: It may be unclear whether all those treatment groups received vaccine administration at the time of receiving therapy. Also, “alone” may be added after “immunotherapy”. I recommend writing like this “within the systemic treatment group, at the time of SARS-CoV-2 mRNA vaccine administration, 14/55 patients were receiving chemotherapy alone, 21/55 were receiving immunotherapy alone, 8/55 were receiving chemotherapy and immunotherapy, and 12/55 were receiving concurrent chemotherapy, immunotherapy, and radiation therapy.”

We thank the reviewer for their suggestion and have modified the sentence as indicated.

(7) Line 171-174: these statements seem to be less relevant to the topic of the paper; the authors may further state the significance of this research observation in the discussion section. 

We have added additional language to the discussion section of our revised manuscript addressing this point.

(8) Line 112: why a duplicate run instead of the more commonly used triplicate run was performed?

For our IFN-g ELISPOT assay we determined that running the peptide simulated wells in duplicate and the negative control (DMSO only) wells in triplicate provided the most efficient use of the assay plate. We have added this information to the Material and Methods section of our manuscript.

(9) Figure 1: please provide an explanation of the meaning of the dotted lines in the caption.

We have added the indicated text to the figure legend.

Minor points:

(10) Table 1, “58.2% 32/55)”, a parenthesis is missing.

 We thank the reviewer for making us aware of this typo.

(11) Reference 14, 17, and 18: page numbers are missing.

We have added the indicated page numbers  

(12) Line 61, line 79, and line 95: keep the formatting of the subtitle consistent (period vs colon vs nothing).

We have standardized the subtitles as suggested

(13) Table 1: some of the numbers are bold numbers and some of them are not. Please provide an explanation or keep the formatting consistent.

We have reformatted the indicated table

Reviewer 2 Report

The authors compared antibody responses between cancer patients with anti-cancer therapy and cancer patients without anti-cancer therapy after receiving second dose of mRNA vaccines. The study could contribute to the current pandemic. I have some comments for the manuscript.

- introduction section: explain why you only focus on certain groups of cancer patients

- line 62: need to specify the study period instead of mentioning the date of initiation

- discussion section:

(1) need to discuss your results with other similar studies

(2) need to describe the limitations of the study, e.g. the duration of the study period, small sample size, unknown previous SARS-CoV-2 infection history

Author Response

- introduction section: explain why you only focus on certain groups of cancer patients

We have endeavored to make our selection criteria clearer in our revised manuscript.

- line 62: need to specify the study period instead of mentioning the date of initiation

We have provided the requested information in the Study Design section of our revised manuscript.

- discussion section:

(1) need to discuss your results with other similar studies

We have added additional language to the discussion section of our manuscript comparing our results to those obtained in other/related studies

(2) need to describe the limitations of the study, e.g. the duration of the study period, small sample size, unknown previous SARS-CoV-2 infection history

We have added a paragraph highlighting these limitations of our study.

Reviewer 3 Report

This manuscript describes a comparison study on SARS-CoV-2 mRNA vaccine-induced immunity, specifically humoral and cellular immunogenicity, between cancer patients undergoing cancer therapies and cancer patients not undergoing any therapy, in order to fill the lack of study into SARS-CoV-2 mRNA vaccine-induced cellular immunogenicity given that the majority of immunogenicity studies on these groups have focused on SARS-CoV-2 spike protein-reactive antibodies. Since many cancer therapies have been associated with causing deficits in patients’ immunity, this study is therefore an important topic to study on the effectiveness of SARS-CoV-2 mRNA vaccines in cancer patients and whether cancer therapy affects this. The results suggest that vaccine-induced cellular immunity develops in both groups, even in cases where there is no significant development of SARS-CoV-2 specific antibodies. This is a very good article and I would recommend it for publication with some potential minor revision (as below):

  • I think the authors should provide a bit more background info of different form of vaccines. For example: International Journal of Biological Sciences 17(6): 1461-1468.
  • Just wonder do the authors also look at the cancer biomarker changes such as CA19.9, CEA etc. before and after the vaccination. I think this is also an important set of data.   

Author Response

  • I think the authors should provide a bit more background info of different form of vaccines. For example: International Journal of Biological Sciences 17(6): 1461-1468.

We have added some additional language to the manuscript highlighting other SARS-CoV-2 vaccines available at the time of study execution. However, please note that we restricted enrollment in this study to individuals who received a SARS-CoV-2 mRNA vaccine.

  • Just wonder do the authors also look at the cancer biomarker changes such as CA19.9, CEA etc. before and after the vaccination. I think this is also an important set of data.

We strongly agree with the reviewer that correlating common cancer biomarkers with vaccine immunogenicity would be interesting and of benefit to the field. However, we did not collect these data as part of this study.